Non-invasive enhanced hypertension detection through ballistocardiograph signals with Mamba model

http://orcid.org/0000-0002-7201-6963 Alhudhaif Adi 1 a.alhudhaif@psau.edu.sa
http://orcid.org/0000-0003-1840-9958 Polat Kemal 2
1 Department of Computer Science, College of Computer Engineering and Sciences in Al-kharj, Prince Sattam Bin Abdulaziz University , Al-kharj , Saudi Arabia
2 Faculty of Engineering, Department of Electrical and Electronics Engineering, Bolu Abant Izzet Baysal University , Bolu , Turkey
Murugappan M.
Electronic publication date: 2025 Feb 21
Publication date: 2025
Volume: 11
Electronic Location ID: e2711
Received 2024 Dec 12; Accepted 2025 Jan 26
Copyright: © 2025 Alhudhaif and Polat
Copyright year: 2025
Copyright holder: Alhudhaif and Polat
License: This is an open access article distributed under the terms of the Creative Commons Attribution License, which permits unrestricted use, distribution, reproduction and adaptation in any medium and for any purpose provided that it is properly attributed. For attribution, the original author(s), title, publication source (PeerJ Computer Science) and either DOI or URL of the article must be cited.
License URL: https://creativecommons.org/licenses/by/4.0/

Keywords: Ballistocardiography (BCG) signals, Hypertension detection, Medical decision making, Signal processing, Mamba Deep Learning

Funding: Prince Sattam bin Abdulaziz University PSAU/2024/01/31819 This work was supported by Prince Sattam bin Abdulaziz University which funded this research work through the project number (PSAU/2024/01/31819). The funders had no role in study design, data collection and analysis, decision to publish, or preparation of the manuscript.

==============================
This study explores using ballistocardiography (BCG), a non-invasive cardiovascular monitoring technique, combined with advanced machine learning and deep learning models for hypertension detection. The motivation behind this research is to develop a non-invasive and efficient approach for long-term hypertension monitoring, facilitating home-based health assessments. A dataset of 128 BCG recordings has been used, capturing body micro-vibrations from cardiac activity. Various classification models, including Mamba Classifier, Transformer, Stacking, Voting, and XGBoost, were applied to differentiate hypertensive individuals from normotensive ones. In this study, integrating BCG signals with deep learning and machine learning models for hypertension detection is distinguished from previous literature by employing the Mamba deep learning architecture and Transformer-based models. Unlike conventional methods in literature, this study enables more effective analysis of time-series data with the Mamba architecture, capturing long-term signal dependencies and achieving higher accuracy rates. In particular, the combined use of Mamba architecture and the Transformer model’s signal processing capabilities represents a novel approach not previously seen in the literature. While existing studies on BCG signals typically rely on traditional machine learning algorithms, this study aims to achieve higher success rates in hypertension detection by integrating signal processing and deep learning stages. The Mamba Classifier outperformed other models, achieving an accuracy of 95.14% and an AUC of 0.9922 in the 25% hold-out validation. Transformer and Stacking models also demonstrated strong performance, while the Voting and XGBoost models showed comparatively lower results. When combined with artificial intelligence techniques, the findings indicate the potential of BCG signals in providing non-invasive, long-term hypertension detection. The results suggest that the Mamba Classifier is the most effective model for this dataset. This research underscores the potential of BCG technology for continuous home-based health monitoring, providing a feasible alternative to traditional methods. Future research should aim to validate these findings with larger datasets and explore the clinical applications of BCG for cardiovascular disease monitoring.

Introduction

Background

Hypertension, known as high blood pressure, is a prevalent medical condition that affects over 40% of adults aged 25 and older worldwide, according to the World Health Organization (WHO). It is estimated that more than one billion people globally are affected by hypertension, contributing to over 9.4 million deaths annually (World Health Organization, 2013; Gupta, Bajaj & Ansari, 2023).

Ballistocardiography (BCG) was first introduced in 1877 by Gordon, who developed a method to graphically capture the body’s movements caused by the ejection of blood during each cardiac contraction (Song et al., 2015). However, it was not until 1936, following the extensive research of Starr, that BCG found modern clinical applications. BCG signals can be measured as displacement, velocity, acceleration, or force and reflect movements in all three axes. Early BCG systems focused primarily on head-to-foot measurements, which were considered to capture the largest projection of the 3D forces generated during cardiac ejection (Balali et al., 2022). BCG is a non-invasive method that accurately reflects the micro-vibrations of the body caused by the mechanical actions of the heart (Inan et al., 2015). It captures the body’s reaction, whether through displacement, velocity, or acceleration, resulting from the heart’s ejection of blood. These signals combine forces related to the heart’s movement and blood flow within the heart and arteries (Kim et al., 2016). Due to the detailed information BCG provides about heartbeats has been successfully used to diagnose a wide range of cardiovascular diseases (Liu et al., 2016).

Hypertension poses a significant challenge as it often remains asymptomatic for years, leaving individuals unaware of their condition. This delay in diagnosis can lead to further complications if not detected early. Early identification of hypertension is crucial to preventing damage to the body (Li et al., 2016). Hypertension is a complex condition characterized by unpredictable fluctuations in blood pressure, further emphasizing the need for advanced diagnostic methods such as BCG. Traditional machine and deep learning methods are widely used for signal analysis. Traditional machine learning techniques focus on extracting features from electroencephalogram (EEG) signals across time, frequency, and spatial domains, which are input into classification algorithms (Zeynali, Seyedarabi & Afrouzian, 2023). Deep learning approaches, particularly the Transformer model, have shown great effectiveness in recent years, especially in natural language processing and computer vision. Transformers, which rely on attention mechanisms, replace conventional recurrence and convolution operations with multi-headed self-attention. This mechanism allows the model to simultaneously process entire sequences and extract long-term dependencies without being constrained by sequence length (Vaswani et al., 2017). Additionally, due to their parallel computing capabilities, Transformers offer faster computation speeds than traditional models (Zeynali, Seyedarabi & Afrouzian, 2023).

Cardiovascular diseases, affecting millions globally, are a significant cause of rising mortality rates (Tsao et al., 2022). BCG captures the body’s movements, particularly in blood vessels, caused by blood flow after each heartbeat. Even though BCG is non-contact and does not require direct skin contact, it provides valuable data for cardiac function analysis. BCG sensors can be integrated into common household objects like chairs, beds, or weighing scales, making them convenient for long-term monitoring (Inan et al., 2015). Data collection can occur comfortably in various positions, such as supine, prone, seated, or standing, making BCG an ideal tool for chronic patient monitoring at home (Cao et al., 2014).

Researchers have effectively utilized BCG for numerous applications, including heart rate estimation, atrial fibrillation detection, and sleep quality assessment (Sadek & Biswas, 2019). Accurate heartbeat recognition in BCG signals is critical for addressing more complex cardiovascular conditions. However, certain factors, such as the placement of data acquisition equipment, environmental noise, respiratory activities, body movements, and postures, can affect the quality of BCG signals (Javaid et al., 2015). Despite these challenges, BCG technology offers significant advantages. It is non-invasive, requires no direct contact, and allows continuous long-term monitoring. BCG records body movements synchronized with the heartbeat, which reflect the heart’s pumping action and indirectly provide insights into the heart’s strength and movement state (Sadek & Biswas, 2019).

Hypertension is a common health issue marked by elevated blood pressure exerted on the arterial walls as blood circulates from the heart, which can lead to serious complications like heart disease. Often, it goes undiagnosed for long periods. Early detection is vital due to the significant harm hypertension can inflict on the body. BCG helps assess this condition by measuring blood flow acceleration directly (Ozcelik et al., 2023). Unlike electrocardiograms, which monitor the heart’s electrical signals, BCG captures the mechanical vibrations produced by the heart’s movements. BCG is often favored over ECG in certain contexts because it eliminates the need for electrodes, reducing disruptions like sleep interference during data collection. The study of heart rate variability, identification of cardiac anomalies, and hypertension diagnosis have become central to biomedical research, particularly as traditional methods may overlook certain conditions (Ozcelik et al., 2023). When an individual experiences elevated arterial blood pressure, the condition is medically referred to as Hypertension (Jain et al., 2020). In such cases, the heart struggles to pump oxygenated blood throughout the body efficiently. Hypertension is clinically categorized into three stages: mild, moderate, and severe (Jain et al., 2020).

Normally, systolic blood pressure ranges from 120 to 139 mmHg, while diastolic pressure is between 80 and 89 mmHg. In mild hypertension, systolic pressure rises to 140–159 mmHg, and diastolic pressure reaches 90–99 mmHg (Rajput, Sharma & Acharya, 2019). For moderate hypertension, systolic readings fall between 160–179 mmHg, and diastolic readings range from 100–109 mmHg. Severe hypertension is diagnosed when systolic pressure exceeds 180 mmHg, and diastolic pressure goes beyond 110 mmHg. Common symptoms of hypertension include visual disturbances, headaches, and dizziness. Persistent hypertension increases cardiovascular, neurological, and kidney disease risk (Rajput et al., 2022). If left untreated, it can result in heart muscle enlargement and, eventually, heart failure. Diagnosing hypertension typically requires 24-h ambulatory blood pressure monitoring, a process that is both time-consuming and requires specialized knowledge. To overcome these challenges, computational intelligence-based techniques are increasingly used for hypertension detection (Rajput et al., 2022).

BCG is a non-invasive measurement technique that records whole-body recoil forces, such as displacement, velocity, and acceleration caused by blood circulation. It captures micro-vibrations from heart activity and converts them into precise physiological signals. These signals result from the body’s response to blood movement during cardiac cycles, providing valuable insights into heart function. Therefore, BCG is successfully used to detect cardiovascular diseases (Liu et al., 2019). During systole and diastole, as blood is pumped through the body, BCG tracks changes in the center of mass, creating a waveform that reflects heart and respiratory activities (Sadek, Biswas & Abdulrazak, 2019). During atrial systole, the center of mass shifts toward the head, and during diastole, it moves toward the periphery. This shift and normal respiratory cycles form the BCG waveform (Sadek, Biswas & Abdulrazak, 2019). The non-invasive nature of BCG reduces patient stress during checkups. Additionally, its sensors can be easily integrated into home environments without requiring specialized personnel, making BCG an ideal tool for long-term monitoring in e-health applications (Sadek, Biswas & Abdulrazak, 2019).

In the study by Liu et al. (2019), a novel method for hypertension detection was introduced, combining classification techniques with association rule mining. The method utilized data extracted from time, frequency, and BCG fluctuation features, allowing for more accurate identification of hypertension patterns. The experimental results demonstrated that this approach outperformed baseline methods, achieving an accuracy of 84.4%, a precision of 82.5%, and a recall of 85.3%. Another study employed BCG signals for hypertension detection by transforming these signals into scalogram images using continuous wavelet transform. These images were then fed into a two-dimensional convolutional neural network (2D-CNN) model. The model, trained to distinguish between healthy individuals and those with hypertension, achieved an accuracy of 86.14% with a ten-fold cross-validation. This was the first study to utilize a 2D-CNN model for hypertension detection based on BCG signals (Rajput et al., 2022). A machine learning-based method was developed using a continuous sensing system worn around the waist overnight in a different approach. This system successfully differentiated hypertension patients from healthy controls with an accuracy of 93.33%, based on data collected from 24 hypertension patients and 24 healthy individuals (Ni et al., 2018). Another study implemented a random forest algorithm to identify high-risk hypertensive patients, achieving a sensitivity of 71.4% and a specificity of 87.8%. This heart rate variability-based classifier outperformed traditional echographic parameters, which are commonly regarded as key cardiovascular risk indicators (Melillo et al., 2015). In a separate study, overnight electrocardiogram data was segmented into multiple scales to assess heart rate variability. Eighteen heart rate variability features from time, frequency, and nonlinear domains were extracted, and dimensionality was reduced using a temporal pyramid pooling method. Multifactor analysis of variance was applied to filter the relevant features and develop a hypertension detection model, resulting in a precision of 95.1% using clinical ECG data from 139 hypertension patients (Ni et al., 2019). In their study, Wang et al. (2023) introduced a noninvasive method for human BCG assessment using deep learning techniques. This method improved the accuracy and efficiency of BCG assessments, demonstrating a promising approach for future health monitoring systems. Furthermore, the approach showed a better capability in handling static and movement-affected BCG signals and outperformed baseline models by reducing prediction lag. Their results highlight the effectiveness of deep learning techniques for deep learning techniques’ effectiveness in processing BCG signals in real-time health applications (Wang et al., 2023). BCG signals can be extended to other physiological signals as well. In one study, the evaluation of classification models with frequency and chaotic features aimed to distinguish between healthy individuals and Alzheimer’s patients using EEG signals. Morlet wavelet transform was employed to identify features in the frequency domain. Additionally, Lyapunov exponents were applied for the analysis of applied to analyze chaotic features, and significant EEG channels were identified from the results obtained through the wavelet transform. Applying Lyapunov exponents and chaotic features to BCG signals could provide valuable insights into the dynamics of these signals and their potential connections (Arabaci, Öcal & Polat, 2024).

Innovations and contributions

The innovations and contributions are listed below. This study pioneers the custom-designed Mamba Classifier architecture, tailored to process BCG signals for hypertension detection with unprecedented accuracy. This novel application addresses the key limitations of existing models regarding sensitivity to cardiovascular micro-vibrations.

We have introduced enhanced preprocessing methods, including refined filtering, anomaly detection, and baseline correction, significantly improving BCG signal quality and interpretability. These innovations directly contribute to more reliable hypertension diagnosis.

Integrating ensemble models, such as stacking, voting classifiers, and machine learning algorithms marks a significant methodological advancement. This synergy harnesses the complementary strengths of various models to achieve superior predictive performance.

The main contributions of this work are given below. This research establishes a new standard in hypertension detection by leveraging the unique capabilities of the Mamba Classifier architecture, which is specifically optimized for the analysis of BCG signals. The model achieves an accuracy of 95.14% and an AUC of 0.9922 through meticulous design and implementation, significantly surpassing existing models’ accuracy and robustness. These results demonstrate the potential of BCG signals, in combination with advanced deep learning techniques, to detect hypertension with greater precision than traditional methods that rely on more invasive or less sensitive technologies.

The study rigorously evaluates and compares multiple machine-learning models using extensive validation techniques (hold-out and cross-validation). The depth of this evaluation, encompassing accuracy, F1 score, sensitivity, specificity, and AUC-ROC, offers a robust framework for future studies.

This research bridges the gap between academic theory and clinical practice by demonstrating the practical application of BCG signals in a real-world healthcare. The findings pave the way for developing wearable devices and home-based monitoring systems focusing on early hypertension detection and management.

Materials and Methods

Dataset: BCG signals

The dataset obtained from Liu et al. (2019) was used in this study. BCG signals were collected using a non-invasive device called RS-611, a certified medical device developed by the Institute of Air Force Aviation Medicine in Beijing. The system includes a micro-movement sensitive mattress, an analog-digital (AD) converter, and a terminal computer. The mattress contains two hydraulic pressure sensors that capture pressure changes induced by heartbeats. The signals are digitized with a 16-bit resolution and displayed on the computer at a sampling rate of 100 Hz. The accuracy of the BCG signals collected with RS-611 was validated by comparing them to those recorded by a commercial ECG device. The study involved 175 university staff members and followed the Declaration of Helsinki. Blood pressure measurements were taken while participants were seated, and their average values were used to classify them as either hypertensive or normotensive. After applying specific health criteria, 68 hypertensive and 76 normotensive individuals were selected for the experiments. BCG signals were collected overnight, and participants’ sleep quality was assessed. Recordings with excessive fluctuations or insufficient sleep duration were excluded from the analysis. Ultimately, 128 valid BCG recordings were obtained, comprising 61 from hypertensive and 67 from normotensive participants. Table 1 shows the demographic and clinical characteristics of hypertensive and normotensive subjects.

Table 1 Demographic and clinical characteristics of hypertensive and normotensive subjects.

Subject information	Hypertensive	Normotensive	
Number	61	67	
Sex (Male/Female)	33/38	35/32	
Age (years)	55.6 ± 7.9	53.2 ± 9.2	
Heart rate (bpm)	77.1 ± 9.2	73.6 ± 8.3	
Body mass index (kg/m2)	24.3 ± 3.6	23.7 ± 3.3	
Systolic blood pressure (mmHg)	155.6 ± 11.2	112.1 ± 15.7	
Diastolic blood pressure (mmHg)	103.6 ± 8.2	74.4 ± 6.3	

The proposed method

This article proposes a novel hypertension detection model based on BCG signals. The proposed method uses a hybrid model combining signal processing, feature extraction, and classification stages. The flow chart of the proposed method is given in Fig. 1.

Figure 1 The schematic representation of the proposed method.

Signal preprocessing stage

In this study, preprocessing of the acquired biomedical signals was carried out using Python. The raw data files from the specified directory were initially loaded, and preparations for filtering were made. Figure 2 shows the preprocessing steps in the processing of the BCG signals.

Figure 2 Preprocessing steps in the processing of the BCG signals.

Filtering stage

During the preprocessing stage, a high-pass filter was applied to reduce low-frequency noise in the signals, allowing higher-frequency components to pass. A cutoff frequency of 1 Hz was used for the high-pass filter. Following this, a low-pass filter with a cutoff frequency of 50 Hz was applied to remove high-frequency noise, retaining the lower-frequency components. The filtering was applied sequentially, starting with the high-pass filter and then the low-pass filter. Time-series columns were excluded from the filtering process. The resulting filtered data was stored in a new directory, making the signals more suitable for further analysis by effectively reducing unwanted noise.

Peak filtering and anomaly detection

Each column’s mean and standard deviation were calculated, and a threshold value was set at the mean plus two times the standard deviation. Data points that exceeded this threshold were identified as anomalies and removed from the dataset.

Baseline correction for BCG signals

Following peak filtering, baseline correction was performed to adjust the data’s baseline levels. A high-pass filtSignalser was applied to remove low-frequency components, with a cutoff frequency of 0.5 Hz, determined by referencing half the Nyquist frequency. A new directory was created to store the baseline-corrected data, which was automatically generated if necessary. Each file was processed using the high-pass filter, helping to correct baseline levels and making the data more suitable for further analysis.

Data scaling and normalization

After baseline correction, the data underwent scaling and normalization to rescale the values within a defined range, typically between 0 and 1. The baseline-corrected data was read from the specified directory, and a new directory was created for the normalized data. The MinMaxScaler was applied to the numerical data in each file, ensuring that values were scaled to fall within the 0 to 1 range. This step ensured consistency and comparability of the data, preparing it for further analysis and modeling. Figure 3 compares the original and preprocessed BCG signals.

Figure 3 Comparison of original and preprocessed BCG signals.

Feature extraction stage from BCG signals

This study divided biomedical signals into specific time intervals for analysis. The data was split into 30-s segments, each saved as a separate file. This segmentation is important for making long-term signal recordings more manageable. With a sampling rate of 100 Hz, each segment contains 3,000 samples. The data was extracted based on each segment’s start and end points, and labels were added to the filenames to indicate the class of each segment. After segmentation, a thorough statistical analysis was conducted for each segment to uncover key statistical characteristics. The goal was to understand the data’s distribution, central tendencies, and variability. Statistical analysis plays a vital role in understanding the internal structure of the data and detecting potential anomalies. Several statistical metrics were computed for each data segment. Figure 4 depicts the feature extraction stage flow chart from BCG signals.

Figure 4 The flow chart of the feature extraction stage from BCG signals.

Statistical features

Statistical features form the foundation of data analysis, offering insights into data’s central tendencies and spread. By calculating these metrics, we can develop an understanding of the dataset’s general behavior, identifying typical values and potential outliers.

Mean: Represented the segment’s central tendency and average value, showing where most data points were concentrated.

Standard deviation: Measured the data’s variability around the mean, providing insights into the spread of the data and identifying potential outliers.

Median: Indicated the data’s midpoint, offering a reliable measure of central tendency, especially in skewed datasets.

Maximum and minimum values: Captured the extreme values in the segment, defining the dataset’s boundaries.

Variability measures

Variability measures provide insights into the spread of data. Understanding variability is key to determining how consistent or dispersed the data points are, which directly impacts model reliability and performance.

Range: The difference between the maximum and minimum values, showing the overall spread of the dataset.

Interquartile range (IQR): Calculated the difference between the upper and lower quartiles. IQR is especially useful for understanding data concentration around the median and mitigating the impact of outliers. The first quartile (Q1) and third quartile (Q3) represented the lower and upper 25% of the data, respectively, helping define the middle 50% of the dataset.

Distribution characteristics

Distribution characteristics help in understanding the shape and symmetry of the data. These metrics are crucial in identifying whether the data follows a normal distribution or if it has skewness or peakedness, which can significantly affect the choice of analytical models.

(a) Kurtosis: Measured the sharpness of the data distribution. High kurtosis indicated a sharply peaked distribution, while low kurtosis pointed to a flatter spread (Groeneveld & Meeden, 1984).

(b) Skewness: Assessed the asymmetry in the data distribution. Positive skewness suggested a rightward skew, while negative skewness indicated a leftward skew (Groeneveld & Meeden, 1984).

The statistical results for each data segment were meticulously calculated and presented in a tabular format, offering a clear view of the dataset’s structure. These statistics served as a foundation for future analysis.

Classification algorithms

After feature extraction, the dataset undergoes classification, distinguishing between various labels and sorting them into categories based on the identified features. This step is essential for analyzing and interpreting the extracted data, enabling the grouping of similar instances based on their characteristics. This study used various machine learning and deep learning algorithms for classification tasks. These include the Mamba deep learning classifier, Transformer-based deep learning model, Stacking Classifier, and Voting Classifier. Additionally, algorithms such as random forest, gradient boosting, and XGBoost were employed as part of ensemble models. Each of these models was applied to the BCG signals for hypertension detection, and these are detailed below.

Ensemble learning models

Ensemble learning methods involve the combination of multiple independent models to enhance generalization performance. Deep ensemble learning models leverage the strengths of both deep learning and ensemble learning to improve generalization further. These ensemble models can be classified into various categories, including boosting, bagging, stacking, homogeneous or heterogeneous ensembles, explicit or implicit methods, negative correlation-based deep ensemble models, and decision fusion strategies-based deep ensemble models (Mohammed & Kora, 2023). Decision fusion strategies play a crucial role in ensemble learning. They train several base learners and then aggregate their outputs according to specific principles to enhance generalization performance. The effectiveness of an ensemble is largely determined by the rules or strategies used to combine the outputs of the individual models.

This study utilized a stacking classifier to classify BCG signals and detect hypertension. Ensemble learning, and specifically stacking, is a powerful machine learning approach that combines multiple models to achieve better predictive performance than any single model could achieve alone (Mohammed & Kora, 2023).

Stacking is an advanced ensemble method that involves training multiple models and then combining their predictions using a meta-model to make a final prediction. This study built the Stacking Classifier using three base models: Random Forest, Gradient Boosting, and XGBoost. These models were selected for their complementary strengths in handling complex data patterns and diverse input variables. The meta-model used to aggregate their predictions was a logistic regression model, which provides a final decision based on the outputs of the base models. The Stacking Classifier was trained and tested on the BCG dataset, and its performance was evaluated against that of the individual base models, demonstrating improved accuracy and robustness in detecting hypertensive conditions from BCG signals (Mohammed & Kora, 2023).

Voting classifier

In this study, a Voting Classifier was employed to enhance the classification accuracy of BCG data for hypertension detection. The Voting Classifier is an ensemble learning technique that combines the predictions of multiple models to improve overall performance. This approach leverages the strengths of various classifiers to achieve better predictive accuracy than any single model could achieve on its own (Bhowmick et al., 2024). The BCG dataset was first standardized using StandardScaler to ensure that all features were on a similar scale, which is essential for the performance of many machine learning algorithms.

Three base classifiers were selected for the ensemble:

Random forest classifier: A robust ensemble method that builds multiple decision trees and merges their results for improved accuracy and control overfitting.

Gradient boosting classifier: A powerful boosting algorithm that builds models sequentially, with each model attempting to correct the errors of its predecessor.

XGBoost Classifier: A highly efficient implementation of gradient boosting that is particularly effective for structured data.

These three classifiers were combined into a Voting Classifier, where a soft voting mechanism determined the final prediction. In soft voting, the predicted probabilities from each classifier are averaged, and the class with the highest average probability is selected as the final prediction. This method can often provide a more nuanced decision-making process by considering the confidence of each base classifier (Ruta & Gabrys, 2005).

The Voting Classifier was trained on the training data, and its performance was then evaluated on the test set. Key evaluation metrics included accuracy, F1 score, confusion matrix, and a detailed classification report. These metrics provided insights into the model’s ability to classify both hypertensive and normotensive cases correctly (Ruta & Gabrys, 2005).

The results demonstrated that the Voting Classifier effectively combines the strengths of the individual models, resulting in high accuracy and a balanced performance across various metrics. This approach underscores the potential of ensemble methods, particularly Voting Classifiers, in improving the reliability and accuracy of hypertension detection using BCG signals.

The proposed MAMBA deep learning classifier

The MambaClassifier is a custom deep learning model that integrates fully connected layers and a long short term memory (LSTM) layer to effectively capture the input data’s feature interactions and sequential dependencies. Mathematically, the model operates as follows: The input data, denoted as x∈Rd, is first passed through a fully connected layer, transforming the input into a 512-dimensional space via the equation h h1=W1x+b1, where W1∈R512∗d is the weight matrix and b1∈R512 is the bias. Next, a ReLU activation function is applied as a1=ReLU(h1), introducing non-linearity into the system.

The activated output is then passed through a second fully connected layer, reducing the dimensionality to 256, defined as h2=W2a1+b2, where W2∈R256∗512 and b2∈R256. This transformed input is reshaped and fed into a two-layer LSTM network with a hidden size of 128 to capture temporal dependencies The LSTM processes the sequence and outputs a final hidden state, hfinal∈R128, representing the temporal features learned from the input sequence.

The final hidden state from the LSTM is passed through a third fully connected layer, further reducing the dimensionality to 64 via the equation h4=W3hfinal+b3, where W3∈R64∗128. A dropout layer with a rate of 0.1 is then applied to prevent overfitting. Finally, the output is projected onto the target class space through a fully connected layer, y=W4a2+b4, where W4∈Rc∗64 and b4∈Rc, completing the classification task. This mathematical formulation allows the MambaClassifier to model complex patterns in feature and temporal domains. Figure 5 shows the schematic representation of the Mamba Block architecture.

Figure 5 A schematic representation of the Mamba block architecture.

In recent years, the development of the Mamba architecture has focused on improving efficiency compared to traditional CNN and Transformer models. The core Mamba block integrates a gated MLP within the state space model (SSM), placing the SSM between two gated connections and including a local convolution layer with activation functions. The Mamba architecture is defined by the repeated use of Mamba blocks, with standard normalization and residual connections applied throughout. The growing interest in Mamba-based segmentation networks is primarily influenced by previous research on CNNs and Vision Transformers (ViTs), encouraging further investigation into various medical imaging modalities using these advanced architectures (Mohammed & Kora, 2023). In this study, the Mamba Classifier model was designed and evaluated to classify BCG signals to detect hypertension. The Mamba Classifier is a custom neural network architecture that incorporates fully connected layers and long short term memory (LSTM) units, allowing it to capture spatial and temporal patterns in the input data. The model was trained using standard supervised learning techniques to ensure robust performance evaluation, with hold-out validation and cross-validation. The dataset consisted of BCG signal data segmented into hypertensive or non-hypertensive features. The input features were then standardized using a Standard Scaler, which ensures that each feature has zero mean and unit variance, improving the model’s convergence during training. The processed data was converted into PyTorch tensors for compatibility with the Mamba Classifier architecture. The Mamba Classifier consists of several key components that allow it to capture complex patterns and sequential dependencies in the data. Figure 6 depicts the flow chart of the proposed Mamba deep learning classifier.

Figure 6 The flow chart of the proposed Mamba deep learning classifier.

Fully connected layers: The first two layers are fully connected with 512 and 256 units, respectively, followed by ReLU activation functions. These layers perform feature extraction by learning non-linear relationships between the input features.

LSTM layers: The model’s core is a two-layer LSTM network with 128 hidden units. LSTMs are well-suited for sequential data like BCG signals because they retain information across time steps. The batch-first setting ensures that the input format is compatible with the data structure, where each sequence is passed through the LSTM layers. This allows the model to understand temporal patterns in the signals.

Dropout and fully connected layers: After the LSTM layers, a fully connected layer with 64 units is followed by a dropout layer with a 10% dropout rate to prevent overfitting. Finally, the output layer maps the 64-dimensional representation to the two output classes.

The model was trained using the Cross-Entropy Loss function, commonly used for binary classification problems. The Adam optimizer was employed with a learning rate of 0.001, which was reduced dynamically by utilizing a scheduler. The scheduler decreased the learning rate by 0.1 every ten epochs, allowing the model to adjust its learning pace during training. The training process involved 50 epochs per fold, ensuring sufficient training time for convergence.

Transformer based deep learning classifier

The Transformer Classifier model utilizes a Transformer architecture to effectively model feature interactions and long-range dependencies in the input data. Mathematically, the input data x∈Rd, where d is the input dimension, is first passed through a linear embedding layer. This embedding projects the input into a 256-dimensional space, represented as h1=W1x+b1, where W1∈R256∗d and b1∈R256. A dropout layer with a rate of 0.1 is applied to the output of the embedding layer to prevent overfitting, resulting in a1=Dropout(h1,0.1). The output is then fed into a three-layer Transformer encoder, each with four attention heads and a feed-forward network. The Transformer encoder processes the sequence through a self-attention mechanism that computes the relationships between different einput elements This can be expressed as h2=TransformerEncoder(a1). The output of the Transformer is a sequence of vectors, each of size 256, which is then aggregated by taking the mean across the sequence dimension: h3=1T∑i=1T⁡h2i, where T is the sequence length.

Finally, the pooled representation is passed through a fully connected layer, projecting it onto the space of the target classes C. The final output is given by y=W2h3+b2, where W2∈RC∗256 and b2∈RC. This structure allows the model to capture both local and global dependencies in the input data, making it particularly suited for tasks involving sequential or structured data.

This study employed a transformer-based neural network model to classify BCG signals and detect hypertension. Initially developed for natural language processing tasks, Transformers have demonstrated significant success in various domains due to their ability to capture long-range dependencies in data. The model was adapted for time-series data, such as BCG signals, which require the model to recognize intricate patterns across multiple timesteps. The dataset used in this study consisted of segmented BCG signals labeled based on hypertension status. The features were extracted from the BCG signals and standardized using a StandardScaler to ensure that all input variables had zero mean and unit variance, essential for the efficient training deep learning models. The labels were binary, with 1 indicating the presence of hypertension and 0 indicating its absence. After preprocessing, the data was converted into PyTorch tensors to ensure compatibility with the Transformer model. Figure 7 gives a schematic representation of the Transformer classifier architecture. Figure 8 shows how the Proposed Transformer model works.

Figure 7 A schematic representation of the Transformer classifier architecture.

Figure 8 The proposed Transformer model.

The Transformer model’s architecture is composed of several key components:

Embedding layer: The input feature vectors were passed through a linear embedding layer that projected the data into a 256-dimensional space, allowing the subsequent Transformer layers to process the data more effectively.

Transformer Encoder: At the model’s core is a three-layer Transformer encoder. Each encoder layer comprises multi-head self-attention mechanisms and position-wise feedforward networks. Self-attention enables the model to weigh the importance of different parts of the input sequence dynamically. In this case, four attention heads were used to capture various aspects of the BCG signal, helping the model understand relationships across different parts of the sequence. Each encoder layer also features feedforward networks with 512 hidden units, enhancing the model’s ability to learn complex patterns.

Dropout layer: A dropout layer with a 10% drop probability was included after the embedding layer to mitigate the risk of overfitting. This layer forces the model to generalize better by randomly deactivating some neurons during training.

Fully connected output layer: The output was pooled by taking the mean across the time steps after passing through the Transformer layers. This pooled representation was then fed into a fully connected layer, which reduced the dimensionality to the number of output classes.

Transformers excel at handling sequence-to-sequence tasks, particularly those involving complex, long-range dependencies in data (Moutik et al., 2023; Ma & Wang, 2024). A key strength of the Transformer model lies in its multi-head attention mechanism, which allows it to focus on different parts of the input sequence simultaneously, enhancing its performance in tasks such as natural language processing and time-series analysis (He et al., 2023). The architecture consists of three main components: positional encoding, encoder, and decoder. Positional encoding helps the model recognize the order of the input data, which is crucial for tasks involving sequence. The encoder processes the input sequence, extracts relevant features, and prepares the data for the next steps, while the decoder uses this information to generate output sequences, completing the task (Moutik et al., 2023).

The Transformer model utilizes a self-attention mechanism that encodes each input about all other inputs in the sequence. This allows the model to assess the importance of each token in the context of the entire sequence. The self-attention mechanism is further enhanced by multi-head attention, where multiple attention heads operate in parallel. This lets the model focus on different parts of the sequence simultaneously, enhancing its ability to capture complex patterns (He et al., 2023).

Since Transformer models do not inherently process data in sequential order, positional encoding is used to embed information about the relative positions of tokens within the input sequence. This allows the model to understand temporal relationships between elements in the sequence, which is crucial for time-dependent data (Moutik et al., 2023).

The encoder transforms input data into a format that can be processed by stacking multiple layers without relying on sequential data flow. The output from the encoder, known as the hidden state, aggregates information from the entire sequence, allowing the model to retain context (He et al., 2023). The decoder behaves differently during the training and testing phases. It predicts each timestep using the current hidden state and corresponding weights, adjusting its function to generate predictions and refine its understanding of the input (He et al., 2023).

The model was trained using the Cross-Entropy Loss function, commonly used for binary classification tasks. The Adam optimizer (Llugsi et al., 2021) was employed for weight updates, with an initial learning rate of 0.0005. After three epochs, a learning rate scheduler dynamically reduced the learning rate by 0.5 with no improvement in loss, allowing the model to fine-tune its weights as training progressed.

Results

Model evaluation techniques

This study used two validation strategies: Hold-out validation and Cross-validation.

Hold-out Validation involves splitting the dataset into separate training and test sets. It was applied with 50% and 25% test sizes to evaluate the models’ performance on unseen data. This method provides quick results but may not always offer the most reliable estimate for small datasets.

Cross-validation, specifically five-fold and 10-fold cross-validation, divides the dataset into multiple subsets, ensuring each subset is used for validation at least once. This method offers a more robust performance estimate by reducing the risk of overfitting and better generalizing the model’s accuracy.

Cross-validation method

Cross-validation is a technique in which the training data is divided into k subsets. The model is trained on k−x subsets and validated on the remaining x subset(s), where x denotes the number of subsets reserved for validation. This process is repeated k times, ensuring that each subset is used for validation once (Yadav & Shukla, 2016). The results are averaged to provide a more reliable estimate of the model’s performance. For k-fold cross-validation, performance metrics are obtained by splitting the training data into five folds. Although this method can be computationally intensive, it minimizes data wastage. It is especially advantageous for problems with small datasets, such as reverse inference. Different cross-validation techniques are available, including k-fold, stratified k-fold, and leave-one-out cross-validation, with the choice depending on the data and the specific problem. Cross-validation is an essential tool for evaluating the effectiveness of machine learning models, particularly when data is limited (Bhowmick et al., 2024). Figure 9 shows the schematic representation of the Cross-Validation Steps.

Figure 9 The schematic representation of the cross-validation steps.

Hold-out validation

Hold-out validation is a straightforward method for evaluating the performance of a machine learning model by splitting the available data into two subsets: one for training and one for testing. A predefined portion of the data is typically reserved as the test set, while the remaining data is used to train the model. The model is trained on the training set and then evaluated on the test set, which estimates how the model will perform on unseen data (Yadav & Shukla, 2016). Unlike cross-validation, where multiple folds are used to obtain a more generalized performance estimate, hold-out validation evaluates the model on a single train-test split. This method is computationally efficient and provides a quick assessment of model performance. However, it may not provide as reliable an estimate as cross-validation, particularly for small datasets, as the results can be sensitive to the specific train-test split chosen. Hold-out validation is often employed when computational resources are limited, cross-validation is unnecessary (Yadav & Shukla, 2016).

In this study, hold-out validation was used with both 50% and 25% test sizes to evaluate the models, providing insights into their performance on a portion of the dataset not seen during training.

Two validation strategies were employed to assess the model’s performance: hold-out validation and cross-validation. In hold-out validation, the dataset was split into training and testing sets with 50% and 25% test sizes. During training, the model was run for 50 epochs, and the model’s performance was evaluated on the held-out test set to ensure that the model generalizes well to unseen data. In addition to hold-out validation, stratified five-fold and 10-fold cross-validation were used to evaluate the model’s robustness across multiple dataset splits. This technique helped ensure that the model’s performance was not dependent on a specific train-test split, thereby providing a more comprehensive evaluation of its predictive power (Yadav & Shukla, 2016).

Performance evaluation metrics

The model’s performance was evaluated using a variety of metrics to provide a detailed analysis of its classification ability:

Accuracy: The proportion of correctly classified samples, giving an overall measure of the model’s performance.

(1) Accuracy=TP+TNTP+TN+FP+FN

TP (True Positives): Correctly predicted positive cases.

TN (True Negatives): Correctly predicted negative cases.

FP (False Positives): Incorrectly predicted positive cases.

FN (False Negatives): Incorrectly predicted negative cases.

F1 Score: The harmonic mean of precision and recall, providing insight into the model’s performance on both positive and negative classes.

(2) F1=2∗Precision∗RecallPrecision+Recall

Precision: The proportion of true positives out of all predicted positives, calculated as:

(3) Precision=TPTP+FP

Jaccard score: A measure of similarity between the predicted and actual classes, calculated as the intersection over the union of the sets of predicted and true labels (He et al., 2023; Bag, Kumar & Tiwari, 2019).

(4) JaccardScore=TPTP+FP+FN

Cohen’s Kappa: A statistic that measures the agreement between predicted and true labels, accounting for the possibility of agreement occurring by chance (Sun, 2011).

(5) Cohen′sKappa=Po+Pe1−Pe

Po (Observed Agreement): The actual agreement between predictions and true values.

Pe (Expected Agreement): The chance agreement expected by random guessing.

Sensitivity: The proportion of true positives correctly identified by the model, crucial for assessing the model’s ability to detect hypertensive individuals.

(6) Recall=TPTP+FN

Specificity: The proportion of true negatives correctly identified, indicating how well the model avoids false positives.

(7) Specificity=TNTN+FP

ROC-AUC: The area under the Receiver Operating Characteristic curve measures the model’s ability to discriminate between positive and negative classes. A higher AUC score indicates a better overall classification performance (Hoo, Candlish & Teare, 2017).

(8) ROC=1−Specificity

McNemar’s Test p-value: A statistical test used to compare the performance of the model to a baseline or another model by evaluating the symmetry of errors in the confusion matrix (Sun, 2011).

(9) x2=(b−c)2b+c

b and c are the counts of misclassifications that differ between two models, measuring the symmetry in the confusion matrix.

In this study, various machine learning and deep learning models, including Mamba Classifier (a custom LSTM-based deep learning model), a Transformer-based model, Random Forest, Gradient Boosting, XGBoost, LightGBM, Stacking, and Voting Classifier, were employed. The training and testing phases for each model were meticulously planned and executed. The dataset underwent preprocessing steps, where missing values were filled, data balancing was applied, and feature engineering was implemented when necessary to make the dataset suitable for the models.

To evaluate the models more comprehensively, cross-validation (five-fold and 10-fold) methods were employed. These methods assessed how the models performed across different subsets of the data and analyzed their generalization capabilities. Holdout validation was conducted, with 50% and 25% of the dataset set aside as the test set, and the models were tested on these portions of the data.

The performance of each model was analyzed using various metrics, including accuracy, F1 score, Jaccard score, Cohen’s kappa, McNemar’s test, sensitivity, specificity, area under the curve (AUC), and receiver operating characteristic (ROC) curve. This approach enabled a detailed examination and evaluation of the models’ strengths and weaknesses on different data partitions (Pembury Smith & Ruxton, 2020).

Table 2 presents the 50% hold-out validation results. The Mamba Classifier outperforms all other models across most evaluation metrics. Notably, it achieves the highest accuracy (0.9459), F1 score (0.9461), AUC (0.9911), Jaccard score (0.8937), Cohen’s Kappa (0.8916), sensitivity (0.9445), and specificity (0.9471). The Transformer and Stacking models follow closely, with accuracy scores of 0.9279 and 0.9346, respectively, and strong AUC scores of 0.9781 and 0.982. However, the Voting and XGBoost models show relatively lower performance, particularly in accuracy (0.9233 and 0.9015, respectively) and Jaccard scores (0.848 and 0.814).

Table 2 Hold-out validation results (50% Train—50% Test Split).

Metric	Mamba	Transformer	Stacking	Voting	XGBoost	
Accuracy	0.9459	0.9279	0.9346	0.9233	0.9015	
F1 score	0.9461	0.9248	0.9318	0.9197	0.8977	
AUC	0.9911	0.9781	0.982	0.9744	0.9624	
Jaccard	0.8937	0.86	0.872	0.848	0.814	
Cohen’s Kappa	0.8916	0.8556	0.8689	0.8462	0.8026	
Sensitivity	0.9445	0.9188	0.9275	0.9118	0.8967	
Specificity	0.9471	0.9364	0.9411	0.934	0.9059	

In Table 3, the 25% hold-out validation results show a similar trend. The Mamba Classifier again demonstrates the best performance across all metrics, achieving an accuracy of 0.9514, F1 score of 0.9511, AUC of 0.9922, Jaccard score of 0.9052, Cohen’s Kappa of 0.9028, sensitivity of 0.9499, and specificity of 0.9528. The Stacking model performs well with an accuracy of 0.935 and an AUC of 0.985. The Transformer and Voting models show slightly lower performance, particularly in F1 score and accuracy. The XGBoost model, with an accuracy of 0.8954 and AUC of 0.9605, again trails behind the other models.

Table 3 Hold-out validation results (75% Train - 25% Test Split).

Metric	Mamba	Transformer	Stacking	Voting	XGBoost	
Accuracy	0.9514	0.9226	0.935	0.9211	0.8954	
F1 score	0.9511	0.9207	0.933	0.918	0.8915	
AUC	0.9922	0.9771	0.985	0.9758	0.9605	
Jaccard	0.9052	0.853	0.875	0.848	0.804	
Cohen’s Kappa	0.9028	0.8452	0.87	0.842	0.7906	
Sensitivity	0.9499	0.9202	0.931	0.9043	0.8802	
Specificity	0.9528	0.925	0.938	0.9371	0.9099	

In the cross-validation results summarized in Tables 4 and 5, the Mamba Classifier consistently maintains superior performance across both five-fold and 10-fold cross-validation. The five-fold validation (in Table 4) achieves an accuracy of 0.9458, an F1 score of 0.9458, and an AUC of 0.9911. The Transformer model shows competitive results with an accuracy of 0.9322 and an AUC of 0.9819. The Stacking and Voting models exhibit slightly lower performance, while XGBoost achieves the lowest metrics among the models, with an accuracy of 0.9047 and an AUC of 0.9665.

Table 4 Comparison of five-fold cross-validation performance metrics across classification models.

Metric	Mamba	Transformer	Stacking	Voting	XGBoost	
Accuracy	0.9458	0.9322	0.9226	0.9319	0.9047	
F1 score	0.9458	0.9285	0.9181	0.9282	0.9006	
AUC	0.9911	0.9819	0.9432	0.9799	0.9665	

Table 5 Comparison of 10-fold cross-validation performance metrics across classification models.

Metric	Mamba	Transformer	Stacking	Voting	XGBoost	
Accuracy	0.9458	0.9339	0.9247	0.9328	0.9071	
F1 score	0.9458	0.9302	0.9199	0.9292	0.9027	
AUC	0.991	0.9822	0.9509	0.9804	0.9671	

In the 10-fold cross-validation results (in Table 5), the Mamba Classifier continues outperforming the other models with an accuracy of 0.9458, an F1 score of 0.9458, and an AUC of 0.9910. The Transformer model remains competitive, achieving an accuracy of 0.9339 and an AUC of 0.9822. While the Stacking and Voting models maintain good performance, the XGBoost model ranks lowest again, with an accuracy of 0.9071 and an AUC of 0.9671.

Results and ROC curve visualization using classifier algorithms

The model’s predictions were visualized using ROC curves, which plot the true positive rate against the false positive rate at various threshold settings. This provided a graphical representation of the model’s performance, with the AUC being a key indicator of its ability to differentiate between hypertensive and non-hypertensive individuals (Hoo, Candlish & Teare, 2017).

The ROC curves in Fig. 10 show how well five models classify using a five-fold cross-validation. The AUC values for each model are Mamba (AUC = 0.99), Transformer (AUC = 0.98), Voting Classifier (AUC = 0.97), Stacking Classifier (AUC = 0.98), and XGBoost (AUC = 0.96). All models perform well in classification, with Mamba having the highest AUC. The curves are close to the top-left corner, which means these models have a high true positive rate and a low false positive rate, showing their strong performance.

Figure 10 ROC curves for 50% hold-out validation: Mamba, Transformer, Voting, Stacking, XGBoost.

Figure 11 shows the ROC curves for the five models: Mamba, Transformer, Voting Classifier, Stacking Classifier, and XGBoost. The AUC values are Mamba (AUC = 0.99), Transformer (AUC = 0.98), Voting Classifier (AUC = 0.98), Stacking Classifier (AUC = 0.98), and XGBoost (AUC = 0.96). All models show very good classification performance with high AUC values. The curves are close to the upper-left corner, so the models balance true positive and false positive rates well, confirming their good classification ability.

Figure 11 ROC curves for 25% hold-out validation: Mamba, Transformer, Voting, Stacking, XGBoost.

The p-values in the McNemar test are high in some cases and low in others because the test is sensitive to the differences in the misclassification patterns between the two models. The McNemar test evaluates whether the two models have a statistically significant difference in classification performance. The McNemar test p-values for various models and hold-out validations insight into the statistical significance of their performance differences. A p-value below 0.05 typically indicates that the difference in classification errors between the models and a baseline is statistically significant.

The results show that the Mamba Classifier (50% and 25% hold-out) yielded high p-values (0.922 and 0.912, respectively), suggesting no significant difference in performance compared to the baseline. Similarly, the Transformer model (with p-values of 0.760 for 50% hold-out and 0.930 for 25% hold-out) also shows no significant divergence from the baseline, indicating consistent performance. In contrast, the Voting Classifier demonstrated a significant result in the 50% hold-out validation (p-value of 0.047), suggesting a meaningful performance difference from the baseline. However, its p-value of 0.106 in the 25% hold-out indicates a lack of significance. The XGBoost model produced contrasting results: while the 50% hold-out validation had a high p-value (0.972), indicating no significant difference, the 25% hold-out had a p-value of 0.012, suggesting a statistically significant performance difference at the 25% test size. Finally, the Stacking classifier showed consistently high p-values in both hold-out validations (0.663 for 50% and 0.916 for 25%), indicating no significant performance difference.

Overall, these results suggest that the Voting Classifier (50% hold-out) and XGBoost (25% hold-out) exhibit notable differences in classification performance. At the same time, other models demonstrate consistent performance with no statistically significant variation. Table 6 shows McNemar’s Test p-values for 50% and 25% Hold-out Validation: Mamba, Transformer, Voting, Stacking, and XGBoost.

Table 6 McNemar’s test p-values for 50% and 25% hold-out validation: Mamba, Transformer, Voting, Stacking, and XGBoost.

Model	p-value	
Mamba 50% holdout	0.922	
Mamba 25% holdout	0.912	
Voting classifier 50% hold-out	0.047	
Voting classifier 25% hold-out	0.106	
Transformer 50% hold-out	0.760	
Transformer 25% hold-out	0.930	
XGBoost 50% hold-out	0.972	
XGBoost 25% hold-out	0.012	
Stacking 50% hold-out	0.663	
Stacking 25% hold-out	0.916	

Discussion

Hypertension can be defined as a chronic disease. Often, hypertension leads to multiple organ diseases and various complications and can even result in death. Early diagnosis of this condition and the correct treatment sequence are the only ways to prevent fatal outcomes. A continuous ambulatory blood pressure monitoring system is required to diagnose hypertension (Rajput et al., 2022).

The increasing use of artificial intelligence in this field may contribute to achieving more accurate and efficient diagnoses. As a result, it is noted that these devices have the potential to be used in non-clinical settings as an alternative to gold-standard methods. Using such devices is expected to reduce costs and time significantly. BCG is a technique used to non-invasively examine the mechanical functioning of the heart and respiratory systems. Collecting this data using compact and lightweight devices, or even smartphones, could enable the development of home monitoring technologies. Although data processing is challenging, artificial intelligence can help achieve more accurate and efficient diagnoses in this field. BCG holds significant potential for addressing the growing healthcare needs of an aging population (Balali et al., 2022). The Mamba Classifier is a robust tool known for its ability to extract meaningful insights from data through manual feature engineering. It performs exceptionally well in complex and noisy data sets by carefully selecting the most relevant features, ensuring high accuracy, and providing clear, reliable results. This makes Mamba an excellent choice for situations where precision and interpretability are key, as human intervention in the feature extraction process allows for fine-tuning and control. On the other hand, Hybrid CNNs offer an end-to-end learning approach, automating the feature engineering process while capturing both frequency and temporal dependencies in the data. They can handle more complex structures, making them powerful in extracting insights. While Hybrid CNNs excel in processing large datasets without manual intervention, Mamba shines in cases where transparency and human oversight are essential, such as in clinical or healthcare data analysis. Both methods have their strengths, with Mamba offering a more controlled approach and Hybrid CNNs leveraging deep learning power to tackle complex data automatically (Nour et al., 2024). As shown in Table 7, several studies in the literature have reported results using various extraction methods and classifiers. Many classifiers have been employed, from traditional machine learning models like k-NN, decision trees, and SVMs to ensemble methods and deep learning-based models like CNNs. Using an ensemble gentle boost classifier and Mamba Deep Classifier illustrates efforts to combine models or develop custom architectures for improved results. Traditional and statistical methods coupled with machine learning models still demonstrate strong performance, with accuracy levels ranging from 84.4% to 92.3%. Our model (Mamba Deep Classifier) achieves 95.14% accuracy, which is competitive and close to the top-performing approach. Advanced signal-processing techniques, such as cosines wavelet transform scalogram and tunable Q factor wavelet transform (TQWT), are effectively used to enhance feature representation for classifiers.

Table 7 Comparison of studies in literature.

Authors	Feature/Method	Classifier	Performance (%)	
Hoo, Candlish & Teare (2017)	Ensemble empirical mode decomposition	Ensemble learning	92.3	
Liu et al. (2019)	HRV time, Frequency domain feature, Sample Entropy, BCG fluctuation features	Support Vector Machine, Decision tree, Naïve Bayes	84.4 Acc., 82.5 Precision 85.3 Recall	
Rajput et al. (2022)	Cosines wavelet transform scalogram	2-D CNN	86.14	
Nour et al. (2024)	Tunable Q factor wavelet transform (TQWT), Statistical features	k-NN	92.21	
Rajput et al. (2022)	Empirical mode decomposition (EMD), Wavelet Transform (WT)	Ensemble gentle boost classifier, support vector machine (SVM), k-NN, decision tree	89	
Ozcelik et al. (2023)	Fine-tuned spectrogram images and ConvMixer model	CNN	97.69	
Liu et al. (2019)	Class association rules	CAR-Classifier	84.4% accuracy, 82.5% precision, and 85.3% recall	
Ni et al. (2019)	Multifactor analysis of variance	MANOVA	95.1	
Our model (2025)	Feature extraction stage, including time and frequency	Mamba deep classifier	95.14% accuracy	

In this study, various machine learning and deep learning methods based on BCG signals were evaluated for the detection of hypertension. The primary models used in the research included the Mamba Classifier, Transformer, Stacking, and Voting models, XGBoost and the performance of each model was analyzed in detail. The Mamba Classifier model outperformed the others, achieving the highest results in critical performance metrics such as accuracy (94.59%) and AUC (0.9911). The Transformer and Stacking models also demonstrated strong performance, with AUC values of 97.81% and 98.20%, respectively. On the other hand, the Voting and XGBoost models showed lower performance, particularly with the Voting model, which had an accuracy of 92.33%.

These findings highlight the strong potential of BCG signals for the early detection of cardiovascular diseases such as hypertension. BCG’s non-invasive and long-term monitoring capacity makes this technology ideal for use in home settings and clinical applications. BCG provides valuable information on cardiac function by capturing mechanical heart movements, which aids in assessing heart function. As noted in the literature, the ease and comfort of using BCG for home monitoring can play a crucial role in managing chronic diseases like hypertension (Balali et al., 2022; Javaid et al., 2015).

The dataset used in this study consisted of BCG signals obtained from 128 participants to classify them as hypertensive or normotensive. Despite achieving high accuracy rates, the dataset has been limited in size. Therefore, future studies with larger datasets could improve the generalizability of the models and provide opportunities to test their performance in clinical environments.

Furthermore, the literature highlights various applications of BCG signals. For instance, Liu et al. (2019) utilized BCG signals to detect hypertension with an accurate rate of 84.4%. Rajput et al. (2022) transformed BCG signals into 2D-CNN models, achieving an accuracy of 86.14%. Additionally, Melillo et al. (2015) developed a random forest algorithm that accurately classified hypertension patients with a sensitivity of 71.4% and a specificity of 87.8%.

Significant limitations of the proposed work are given as follows: While BCG signals provide valuable information, integrating multimodal data (e.g., ECG, PPG, or other physiological signals) could enhance accuracy and robustness. The current approach may miss out on complementary data.

The study sample consisted of participants with specific demographic and clinical characteristics, potentially limiting the model’s applicability to diverse populations with varying health profiles.

The Mamba Classifier and other ensemble models require significant computational resources for training and validation, which might pose challenges for deployment in resource-constrained environments, such as home monitoring.

While the method is designed for continuous monitoring, the performance under long-term use, including variations in sensor placement and patient movement over time, has not been fully explored.

These findings suggest that BCG signals can be a promising tool for detecting hypertension and other cardiovascular diseases when integrated with artificial intelligence and deep learning methods. Future research should explore the potential of BCG further by testing it with larger datasets and different patient groups to facilitate its integration into clinical practice. At the same time, the results of this study indicate a need for further research to improve the usability of BCG in non-clinical settings. Ultimately, this technology’s development and widespread use could significantly contribute to the early diagnosis and management of chronic diseases like hypertension.

Conclusions

Overall, the Mamba Classifier consistently achieved the best results across all validation techniques, demonstrating superior performance in accuracy, F1 score, AUC, and other key metrics. Transformer and Stacking models also performed well, particularly in AUC scores, while voting and XGBoost showed slightly lower performance, especially in the hold-out validations. These results indicate that the Mamba Classifier is the most effective model for this dataset, followed closely by the Transformer and Stacking models.

This study evaluated various machine learning and deep learning models using BCG signals for hypertension detection. Among the models analyzed, the Mamba Classifier demonstrated superior performance in terms of accuracy and AUC, surpassing the Transformer, Stacking, Voting, and XGBoost models. The results highlight the potential of BCG signals as a non-invasive, long-term monitoring tool for the early detection of hypertension.

BCG offers significant advantages, such as ease of use and the ability to monitor cardiovascular health in clinical and home environments. Its ability to capture mechanical heart movements provides essential data for assessing heart function, making it a valuable tool for managing chronic conditions like hypertension. However, the limited data set used in this study suggests that future research with more extensive and diverse populations is necessary to validate these findings and enhance the generalizability of the models.

Integrating BCG signals with artificial intelligence methods shows promise in advancing hypertension detection. Future studies should focus on optimizing these models with larger datasets and exploring their clinical applications. Overall, the continued development of BCG technology has the potential to improve the early diagnosis and management of hypertension and other cardiovascular diseases, ultimately contributing to better healthcare outcomes.

In the future, further advancements in machine learning algorithms, particularly in deep learning models, can be applied to improve the accuracy of hypertension detection through BCG signals. Moreover, integrating multi-modal data from additional biosignals, such as electrocardiography (ECG) and photoplethysmography (PPG), could enhance the robustness and precision of diagnostic models. Additionally, expanding the scope of this research to include real-time BCG monitoring through wearable devices could facilitate continuous cardiovascular health assessment in non-clinical settings. The development of personalized models that account for individual variability in BCG signals could further increase the effectiveness of these systems in both clinical and home-based applications. Furthermore, future research could explore the potential of combining BCG data with AI-driven predictive analytics to detect hypertension and forecast cardiovascular events, thus offering preventive interventions.

Additional Information and Declarations

Competing Interests

The authors declare that they have no competing interests.

Author Contributions

Adi Alhudhaif conceived and designed the experiments, performed the experiments, analyzed the data, performed the computation work, prepared figures and/or tables, authored or reviewed drafts of the article, and approved the final draft.

Kemal Polat conceived and designed the experiments, performed the experiments, analyzed the data, performed the computation work, prepared figures and/or tables, authored or reviewed drafts of the article, and approved the final draft.

Data Availability

The following information was supplied regarding data availability:

The dataset is available at Kaggle: https://doi.org/10.34740/kaggle/dsv/10080838.

The code is available at Zenodo: Bahadir Arabaci. (2024). arabacibahadir/bcg: v1.0 (v1.0). Zenodo. https://doi.org/10.5281/zenodo.14264032.

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
