# Peer review of "Non-invasive enhanced hypertension detection through ballistocardiograph signals with Mamba model"

_PeerJ Computer Science, doi:10.7717/peerj-cs.2711_

## Round 0.1 · original submission · Major Revisions

I strongly encourage the authors to review and address the reviewer(s) comments carefully. Some comments are critical and related to the need to conduct additional experiments and benchmarking.

Reviewer 1 ·

Basic reporting

-- The aim of the study is to develop a non-invasive and efficient approach for long-term hypertension monitoring, facilitating home-based health assessments.
-- The study is a study that contributes to literature in general. However, the following corrections are recommended.
-- The text should be revised in terms of English language.
-- There is an error in the use of abbreviations. It should be corrected.
-- References should be given in the same standard in the text. Some are capitalized and some are lowercase.

Experimental design

-- Figures and texts are given after the article. Tables and figures should be given in the first place after reference.
-- The introductory part of the study should be improved. It should be written in more detail. The literature review should be expanded.

Validity of the findings

-- Tables and figures are given in appropriate logical order.

Additional comments

-- There are many errors in the study. Errors need to be revised. It should be re-read in English, and references should be edited.

Reviewer 2 ·

Basic reporting

In this manuscript, authors apply the Mamba to monitor human hypertension with ballistocardiography (BCG) signal processing. There are some concerns that need to be addressed by authors.

1. Title: The title needs to be revised. A title something like the following might be more appropriate: "Enhanced hypertension detection through non-invasive ballistocardiography with Mamba"?
2. The quality of figures is really poor from both the perspective of aesthetics and information, the reviewer can not read the text clearly. Authors should revise it carefully.
3. In experiment, authors should add some latest methods for comparison to prove the progressiveness of the Mamba model.
4. It is recommended to conduct experiments based on more other public BCG datasets to verify the correctness and validity of the model.
5. It would be beneficial to further discuss the limitations of the study and potential future directions in more depth. A possible point could be "Non-Invasive Human Ballistocardiography Assessment Based on Deep Learning", https://ieeexplore.ieee.org/document/10121645.

6. A more detailed comparison with other state-of-the-art models, including their advantages and disadvantages, could enhance the significance of the Mamba model.
7. The manuscript needs to be proof-read for better readability and proper grammar. There are some grammar problems in this manuscript.

Experimental design

no comment

Validity of the findings

no comment

---

## Round 0.2 · Major Revisions

I appreciate the author's effort to address the reviewer's comments. Still, one of the reviewers has given suggestions for further improvement.

In addition, there are a few queries from my side to be addressed during the revision.

1. The authors mentioned using the open-source data available in Reference [17] for their work. In this case, how the authors have created a separate link for sharing the same dataset in different places?. Its not fair. Since the dataset is already available in the cloud, its preferred to include the link instead of creating your link.

2. The performance comparison reported in Table 7 does not include the performance of the proposed work. Include the performance of your work and compare the strengths and weakness of your method in contrast with other existing work and discuss why your proposed methodology achieves high/low accuracy compared to other works.

3. Include the major limitations of this present work.

Reviewer 1 ·

Basic reporting

no comment

Experimental design

no comment

Validity of the findings

no comment

Reviewer 2 ·

Basic reporting

1. The title should be changed as "Non-invasive Enhanced Hypertension Detection through Ballistocardiograph Signals with Mamba Model", which is much more suitable and readable.
2. The reviewers hope that the authors will improve the Mamba model in future works to verify the powerful effect of the Mamba model in human healthcare monitoring.

Experimental design

no comment

Validity of the findings

no comment

---

## Round 0.3 · accepted · Accept

The authors have addressed all the major concerns the reviewer(s) and editor raised. The paper can be accepted now.